# Humic Acids Affect the Detection of Metal Ions by Cyanobacteria Carbon Quantum Dots Differently

**DOI:** 10.3390/ijerph191610225

**Published:** 2022-08-17

**Authors:** Simin Liu, Yishen Shi, Xiaona Li, Zhenyu Wang

**Affiliations:** 1Institute of Environmental Processes and Pollution Control, School of Environment and Civil Engineering, Jiangnan University, Wuxi 214122, China; 2Jiangsu Engineering Laboratory for Biomass Energy and Carbon Reduction Technology, Jiangnan University, Wuxi 214122, China

**Keywords:** cyanobacteria, dissolved organic matter, fluorescence quenching, metal ions, natural aquatic environment

## Abstract

A “top-down” synthesis of carbon quantum dots (CQDs), novel fluorescent C materials from waste biomass, is both cost-effective and environmentally friendly. N-rich cyanobacteria are promising precursors to produce CQDs with high fluorescence (FL) intensity for the detection of metal ions. Herein, we synthesized cyanobacteria-based CQDs using a hydrothermal process and evidenced their high FL intensity and stability. The cyanobacteria-based CQDs showed powerful sensitivity for the specific detection of Fe^3+^ and Cr^6+^, which could be ascribed to (i) static FL quenching as a result of the interaction between –OH, –NH_2_, and –COOH groups with the metal ions, (ii) internal filtering effects between the CQDs and Fe^3+^ or Cr^6+^, and (iii) fluorescence resonance energy transfer between CQDs and Cr^6+^. Humic acids (HAs) coexisting led to an underestimation of Fe^3+^ but an overestimation of Cr^6+^ by the CQDs due to the different FL quenching mechanisms of the CQDs. HAs sorbed Fe^3+^ and wrapped the CQDs to form a barrier between them, inhibiting FL quenching of CQDs by Fe^3+^. As for Cr^6+^, HAs reduced Cr^6+^ and also led to FL quenching; the sorbed HAs on the CQDs acted as a carrier of electrons between Cr^6+^ and the CQDs, enhancing FL quenching of the CQDs. This study is the first work to evidence the interference of HAs in the detection of metal ions by CQDs derived from cyanobacteria, which would enlighten the application of CQDs in a natural aqueous environment.

## 1. Introduction

Carbon quantum dots (CQDs) are nanosized fluorescent C materials with a diameter less than 10 nm [1]. They are generally hydrophilic, biocompatible, highly photostable, abundant in surface functional groups, and unique in an electro-chemiluminescence property [2]. CQDs are widely used in biological imaging [3], optical sensors [4], nanomedicine [5], wastewater treatment, and the detection of organic additives or metal ions [6,7]. A “top-down” method for CQD production is to cut down the natural C macromolecules into nanoparticles through physical approaches, including hydrothermal processes, which opens a new avenue for the renewable utilization of a wide range of waste biomass C [8]. A “top-down” method for hydrothermal synthesis of CQDs is both cost-effective and environmentally friendly.

Cyanobacteria are one of the main culprits of eutrophication, a serious environmental issue in aqueous environments. The development of feasible methods for recycling and reusing widespread cyanobacteria biomass is critical to addressing this issue. During the exploration of new C precursors for green synthesis of CQD, cyanobacteria have attracted increasing attention because they are phototropic and nitrogen (N)-rich [9]. It is well known that N-doped functional groups could introduce many surface defects and induce sp^2^ hybrid sites on CQDs [10], which likely improves the yield and fluorescence (FL) intensity of CODs [11,12,13]. However, the incorporation of N into CQDs requires specific chemical and experimental conditions, suffering from a relatively high cost. Therefore, using N-rich waste biomass such as cyanobacteria as feedstock to produce N-doped CQDs is the most economical and simple procedure [10]. Recycling the C from cyanobacteria for CQD production is expected to be of importance for both the development of CQD synthesis technology and environmental sustainability.

Mental ions such as Cr^6+^ and Fe^3+^ are industrial products introduced into water environments via rain off or runoff [14,15]. Exposure to metal ions poses heavy threats to environmental health [16,17,18]. It was reported that Cr^6+^ exposure significantly shortens the lifespan of fish and increases the risks of respiratory diseases and lung cancer in humans [19]. The detection of metal ions is of great concern for the assessment of the environmental health issues they pose. CQDs are promising candidates for the detection of metal ions since the FL properties of CQDs could be changed with the specific binding of metal ions and functional groups on CQDs [20]. For example, Fe^2+^ interacts with the –OH, –NH_2_, and –COOH groups on the surface of CQDs derived from mango leaves, resulting in decreased FL intensity of the CQDs, and the decreased FL intensity is in line with the concentration of Fe^2+^ [21]. In addition, the overlap between the emission spectrum of CQDs and the absorption spectrum of metal ions causes internal filtering effects (IFE), addressing the detection of metal ions [22]. The FL responses of CQDs to metal ions and associated interaction mechanisms depend on the raw materials of CQDs and specific metal ions. CQDs derived from cyanobacteria contain abundant N-doped structures, enhancing light absorption and energy levels of the CQDs and thus being potentially sensitive to metal ions [20]. However, at present, the CQDs produced from cyanobacteria are mostly used in biological imaging and fluorescent films rather than in the detection of metal ions in aqueous environments [9]. Therefore, which metal ions could be specifically detected by the green CQDs derived from cyanobacteria and the associated mechanisms are still unknown.

When CQDs are applied to detect metal ions in natural environments, coexisting dissolved organic matter provides one of the main interferences since it can combine with metal ions or CQDs [23,24]. For example, humic acids (HAs), one of the main components of dissolved organic matter, could combine with Fe^3+^ via strong chemical bonds [25]. A large number of groups such as –OH and –C=O contained in the HAs facilitate their complex physicochemical interactions with other coexisted ions [25,26]. Moreover, HAs can also produce FL under specific excitation conditions [27], which may overlap with that of CQDs or metal ions in a wide spectral range. Therefore, the coexisted HAs are likely to be a bottleneck for accurate FL detection of metal ions by CQDs technology in natural water. However, there are few reports regarding the effects of HAs. To date, how HAs affect the interaction between CQDs and metal ions is still unknown.

This study attempts to propose a green synthesis strategy to generate highly fluorescent and stable CQDs using a low-cost and N-rich cyanobacteria biomass via a green hydrothermal heating process. We hypothesize that abundant N-doped surface functional groups could be formed on the surface of CQDs, contributing to their efficient detection of metal ions; moreover, dissolved organic matter would interfere with the detection of metal ions by the CQDs. The synthesized CQDs were employed to react with various metal ions to determine the specific detection of metal ions and the associated FL response mechanisms affected by dissolved organic matter. The results will pave a new road for the development of CQD synthesis for the effective detection of metal ions and enlighten the application of CQDs in a natural aqueous environment.

## 2. Materials and Methods

### 2.1. Materials and Chemicals

Cyanobacteria were collected from Taihu, Wuxi, China. FeCl_3_ (99%). MgCl_2_·6H_2_O (98%) was purchased from Adamas-beta Co., Ltd. (Shanghai, China) and Shanghai Aladdin Reagent Co., Ltd. (Shanghai, China) K_2_CrO_7_ (99.8%), FeCl_2_·4H_2_O (99%), CuSO_4_·5H_2_O (99%), MnCl_2_·4H_2_O (98%), ZnSO_4_·7H_2_O (99.5%), and Hg(NO_3_)_2_·H_2_O (97.5%) were purchased from Sinopharm Chemical Reagent Co., Ltd. (Shanghai, China) Deionized water, purified using a Millipore water purification system, was used in all the experiments. The HAs used in this study are pH 6.57, containing –COOH, –OH, –NH, and –C–O functional groups [28].

### 2.2. Synthesis of the CQDs

Cyanobacteria were cleaned with deionized water and then oven-dried at 60 °C for 10 h to remove moisture. The dried cyanobacteria were pulverized and passed through a 100-mesh stainless sieve for the following one-step autoclave-assisted hydrothermal process [29]. Briefly, 5.0 g of cyanobacteria were dispersed into 250 mL deionized water with stirring, and the solution was transferred into a 300 mL autoclave equipped with a polytetrafluoroethylene inner wall. The autoclave was heated at 180 °C for 10 h of reaction. After the reaction, the autoclave was cooled to room temperature, and the solution was centrifuged at 8000 rpm for 10 min, filtered with a 0.22 μm polytetrafluoroethylene filter (Jet Bio-Filtration Co., Ltd., Guangzhou, China), and then purified with a dialysis membrane with a molecular weight of 1 KDa for 48 h to remove unreacted small molecules to obtain pure CQDs. Finally, the purified CQDs were vacuum freeze-dried and dissolved in deionized water with a concentration of 100 mg L^−1^ for characterization and experiments.

### 2.3. Stability and pH-Dependent FL Intensity of the CQDs

The optical absorption spectrum of the CQDs in wavelengths ranging from 200 to 700 nm was obtained using an Ultraviolet-Visible (UV-VIS) spectrophotometer (UV-1800, Shimadzu, Kyoto, Japan). The optimal excitation and emission wavelength of CQDs in wavelengths ranging from 280 to 400 nm and 300 to 700 nm, respectively, were determined using an FL spectrophotometer (F-7000, Hitachi, Chiyoda City, Japan), and the bandpass for excitation and emission was set as 10 nm. The pH-dependent FL intensity of the CQDs was plotted against pH values at 4.5, 5.0, 5.5, 6.0, 6.5, 7.0, 8.0, and 9.0, respectively, with the pH of the solution being adjusted using NaOH or HCl. The FL stability of the CQDs was studied under room temperature conditions for 240 min.

### 2.4. Selectivity of the CQDs to Different Metal Ions

To determine the selectivity of CQDs to different common metal ions, including Cr^6+^, Fe^2+^, Fe^3+^, Cu^2+^, Mn^2+^, Pb^2+^, Zn^2+^, Hg^2+^, and Mg^2+^, and the effects of HAs on the detection, the UV-VIS absorption spectra of CQDs, HAs, and different metal ions (with a concentration of 50 mg L^−1^) were measured using a UV-VIS spectrophotometer (UV-1800, Shimadzu, Kyoto, Japan). The most sensitive metal ions to CQDs were determined according to whether there are IFE and fluorescence resonance energy transfer (FRET) mechanisms between the metal ions and CQDs. Further batch experiments were performed with a solution of these sensitive targeted metal ions at an optimal pH value. The corresponding experiment was conducted under optimal excitation and emission wavelength of CQDs, and the comparable experimental condition was at 25 °C, shaken at 150 rpm min^−1^ for 5 min.

### 2.5. Effects of HAs on the FL Intensity of CQDs and the Metal Ions Detection

To determine the effects of HAs on the FL intensity of the CQDs with an excitation at 360 nm, a plot with the FL intensity of CQDs against HAs concentration in a range of 0, 0.1, 0.5, 1, 2, 5, 10, 20 mg L^−1^ was conducted, suggesting concentration-independent effects of HAs on the FL intensity of the CQDs (Appendix A). Concentrations of targeted sensitive metal ions were set as 0, 1, 5, 10, 25, 50, and 100 mg L^−1^, respectively, and the metal ions were determined by CQDs with or without HAs in a 2 mL solution. The bottles were shaken at 150 rpm min^−1^ for 5 min (the timepoint when HAs begin reducing the FL intensity of CQDs (Appendix A), and then the corresponding FL emission spectra were recorded under 360 nm excitation. The limit of detection (LOD) of the CQDs to targeted sensitive metal ions in the solution with or without HAs coexisting was calculated as follows:(1)$$\mathrm{LOD}=\frac{r\times SD}{k}$$
where *r* represents the signal-to-noise ratio of the instrument, which is 3 in default; *SD* represents the standard deviation of the FL intensity of the cyanobacteria CQDs with or without HAs coexisting; *k* represents the fitted slope of the linear line that *F*_0_/*F* against the corresponding concentration of the metal ions, where *F*_0_ is the initial with or without HAs coexisting and *F* is the FL intensity of the cyanobacteria CQDs after interaction with metal ions.

### 2.6. Characterization of the CQDs

The shape, particle size, and surface morphology of the CQDs with or without the effects of HAs were determined using a transmission electron microscope (TEM) which was equipped with JEM-2100 transmission (JEM-2100, Jeol, Akishima, Japan) and operated at an accelerating voltage of 200 kV [9]. The chemical bonds on the CQDs were characterized using a Fourier-transform infrared spectrometer (FTIR) (IRTracer-100, Shimadzu, Kyoto, Japan), with the recorded spectrum in the range of 600 cm^−1^ and 4000 cm^−1^. The oxidation state and functional groups of the CQDs were determined using X-ray photoelectron spectroscopy (XPS) (Thermo Kalpha, Thermo Fisher Scientific, Waltham, MA, USA) and peaked using XPS Peak v.4.1 [9].

## 3. Results and Discussion

### 3.1. Physicochemical Properties of the Cyanobacteria CQDs

The cyanobacteria CQDs are irregular spheres with particle sizes in a range of 1.9 and 6.1 nm and an average diameter of 3.3 nm (Figure 1A). FTIR spectra of the CQDs suggested that many functional groups were formed on the surface of CQDs (Figure 1B). The absorption peak at 3190.0 cm^−1^ could be ascribed to the stretching mode of O–H and N–H [30]; the peaks at 2875.4, 2962.7, and 3056.9 cm^−1^ correspond to the C–H stretching vibrations [9,31]. The absorption peaks observed at 1653.2, 1576.5, and 1067.8 cm^−1^ reflect the stretching of C=C, C=O, and C–O bonds, respectively [9]. Additionally, the absorption peaks located at 1405.6 cm^−1^ and 1297.4 cm^−1^ are assigned to the stretching vibrations of C–N [9]. Functional groups on the surface of CQDs play important roles in their FL emission [20]. For example, protonation of functional groups such as O–H and C–O benefited the high FL intensity of CQDs under acidic conditions [9,29]. However, in a strong acidic condition, a large number of –COOH groups occupied the surface of CQDs, inhibiting the recombination of electron-hole pairs and thus resulting in a declined FL intensity of CQDs [32]. The steric hindrance effect caused by these groups on the surface of CQDs is also one of the main mechanisms that CQDs generally have under weak FL intensity in a strongly acidic or basic environment [28]. Cyanobacteria are eutrophic, resulting in the enrichment of N-containing groups on the CQDs (Figure 1B). Moreover, hydrophilic groups such as O–H, N–H, and C=O and hydrophobic groups such as C=C suggested that the cyanobacteria CQDs are amphiphilic, contributing to their high water-solubility and biocompatibility [33].

The XPS results further determined that the CQDs were mainly composed of three typical characteristic binding peaks of C (C1s, accounting for 66.88%), N (N1s, accounting for 7.57%), and oxygen (O1s, accounting for 25.55%), which were observed at 285.1 eV, 400.1 eV, and 532.1 eV, respectively (Figure 1C). There were no impurities such as metal oxides or salts contained in the CQDs, suggesting that they are environmentally friendly materials and suitable for the detection of metal ions. The high-resolution C1s spectra were fitted by four peaks corresponding to C–C/C=C at 284.5 eV, C–N at 285.0 eV, C–O at 286.2 eV, and C=O at 287.7 eV (Figure 1D) [9]. The most abundant C–N structure, which accounted for 32.29%, could be ascribed to the protein components in cyanobacteria. The C=C infers a graphene structure of the CQDs [34]. The high proportion of O, existing in the form of C–OH/C-O-C at 532.2 eV and C=O at 531.1 eV (Figure 1F), in the CQDs was indicative of a sufficient oxidation process during their production [9]. There was high N content in the CQDs as a result of the fact that cyanobacteria are nutrient-rich [11]. The high-resolution N1s spectra were fitted by three peaks corresponding to C–N=C at 399.4 eV, N–(C)_3_ at 400.1 eV, and N-H at 401.1 eV (Figure 1E) [29]. Compounds with a N–(C)_3_ structure have been widely used as surfactants. The diverse N-dropped structure suggested that protein, nucleic acids, or amino acids, which could contribute to the occurrence of redox reactions on CQDs, are likely produced during the hydrothermal processes of cyanobacteria [35]. Consistent with the FTIR spectrum, the XPS spectrum also indicated that the cyanobacteria CQDs were amphiphilic, conducive to their wide application.

An optical absorption peak of the cyanobacteria CQDs was around 250 nm (Appendix A), which could be ascribed to a π→π* transition of C=C functional groups on the C materials [22]. The FL intensity of the CQDs had a typical excitation-dependent emission behavior (Appendix A). Specifically, with the increased excitation wavelength from 280 to 400 nm, a redshift occurred in the FL intensity peak. The maximum FL intensity of the CQDs was finally obtained under 360 nm excitation and 430 nm emission wavelength (Appendix A), showing a strong blue fluorescence (Appendix A). Generally, the ground state electrons on the CQDs could absorb enough energy, which can be excited under photoluminescence. The FL intensity of the produced CQDs was stable at room temperature for more than 4 h (Appendix A), indicating that there was no FL attenuation of the cyanobacteria CQDs during the detection of metal ions in this study. Moreover, the FL intensity of the produced CQDs depended on pH values in the range of 4 to 9 (Appendix A). Effects of pH on FL intensity of CQDs can be ascribed to protonation and deprotonation of functional groups such as C=O, O–H, and various N-doped functional groups on the surface of CQDs [36,37,38]. In alkaline conditions, the fluorescence intensity of CQDs can be ascribed to the oxidation of C-O functional groups [39]. In this study, the strongest FL intensity of CQDs was in the solution with a pH of 4.5 (Appendix A), which should be suitable for use in the detection of metal ions in industrial wastewater [40]. Our following experiments for the detection of metal ions by CQDs were conducted at pH 4.5.

### 3.2. FL Quenching of the CQDs by Different Metal Ions and HAs

IFE and FRET are two mechanisms of FL quenching of CQDs for the detection of metal ions. Spectral overlaps between the absorption of the acceptor molecule, such as metal ions and the excitation spectrum of the donor molecule, such as CQDs, result in IFE [41]. The IFE effect has been used as an effective strategy for the analysis of the conversion between absorption and fluorescent signals [22,42]. In addition, a FRET phenomenon occurs when the emission spectrum of the donor molecule overlaps with the absorption spectrum of the acceptor molecule in a big area [43], which also contributes to the sensitivity of CQDs to metal ions. As shown in Figure 2A, the excitation spectra of CQDs specifically overlap with the absorption spectra of Cr^6+^ and Fe^3+^ in a large area, while only a small overlap was observed with the absorption spectra of Hg^2+^, Cu^2+^ and Pb^2+^, and no overlap was determined with the absorption spectra of Fe^2+^, Mn^2+^, Zn^2+^, and Mg^2+^. This suggests that cyanobacteria CQDs are sensitive to the detection of Cr^6+^ and Fe^3+^ ions, showing significant FL quenching of the CQDs by the two ions with an IEF mechanism. The Cr^6+^ showed broad absorption peaks at 260 and 350 nm, and Fe^3+^ showed an absorption peak at 300 nm. Except for Cr^6+^, the absorption spectra of the other eight ions do not overlap with the emission spectra of CQDs (Figure 2B). This suggests that the cyanobacteria CQDs could also interact with Cr^6+^ and result in FL quenching of CQDs based on a FRET mechanism. The metal ions with broad absorption spectra and high extinction coefficients generally act as FL quenchers for FRET acceptors [44]. Since the cyanobacteria CQDs showed strong detection properties both for Fe^3+^ and Cr^6+^, it is necessary to add specific blending agents to discharge the interference of another metal ion for concentration detection. Whether there is a superposition effect on the FL quenching of CQDs by the two metal ions deserves to be investigated further. Moreover, an overlap between the absorption spectrum of HAs and the excitation spectrum but not the emission spectrum of CQDs was also observed (Figure 2). This is because CQDs can interact with HAs by IFE but not FRET [23]. The detection of CQDs for metal ions is likely to be affected by the HAs coexisting via IFE effects interference. When the CQDs were applied in the detection of metal ions in natural water, the effects of coexisted metal ions and dissolved organic matter were inevitable.

### 3.3. Detection of Fe^3+^ and Cr^6+^ by CQDs and the Effects of Coexisted HAs

The FL intensity of CQDs against wavelength in the solution contaminated with Fe^3+^ or Cr^6+^ suggested that the FL quenching degree of the CQDs increased with the increased concentration of Fe^3+^ (Appendix A) and Cr^6+^ (Appendix A).

Regardless of HAs coexisting, there was a significant positive linear relationship between the ratio of FL intensity (*F*_0_/*F*) of CQDs and Fe^3+^ with the concentration of metal ions in the range of 0–100 mg L^−1^ (Figure 3A). It was the complexation between CQDs and Fe^3+^ that resulted in electrons transfer from the CQDs to Fe^3+^ and a resultant static FL quenching of the CQDs [45]. The –COOH and –OH functional groups on the CQDs mainly contributed to interacting with Fe^3+^ [46]. The LOD of the cyanobacteria CQDs to Fe^3+^ was 0.6026 mg L^−1^ with a high regression coefficiency (R^2^) of 0.999. Another study produced CQDs from critic acids using a “down-top” synthesis method and suggested that the CQDs showed outstanding Fe^3+^ detection with LOD of 1.0280 mg L^−1^ [47]. The coexisted HAs resulted in an underestimation of the concentration of Fe^3+^ in the solution (Figure 3A). Moreover, with the HAs coexisting, the R^2^ value between the *F*_0_/*F* and Fe^3+^ reduced from 0.999 to 0.960, indicating that HAs interfered with the stable detection of Fe^3+^ by the cyanobacteria CQDs.

Different from the interaction of CQDs to Fe^3+^, there was a significantly binary linear relationship between the ratio of *F*_0_/*F* of CQDs and Cr^6+^ with the concentration of metal ions in the range of 0–100 mg L^−1^ (Figure 3B). A linear relationship between the *F*_0_/*F* of CQDs and Cr^6+^ was only observed when the concentration of Cr^6+^ was lower than 5 mg L^−1^ (Figure 3B). The IFE and FRET between the CQDs and Cr^6+^ contributed to the FL quenching of the CQDs (Figure 2). In addition, Cr^6+^ could be easily reduced into Cr^3+^ or Cr^0^ by reducing functional groups such –O–H and –N–H on the cyanobacteria CQDs (Figure 1B), contributing to FL quenching of CQDs. The LOD of the cyanobacteria CQDs to Cr^6+^ was 0.6831 mg L^−1^ with R^2^ = 0.981. Compared to Cr^6+^, the cyanobacteria CQDs were more sensitive to Fe^3+^ (LOD = 0.6026 mg L^−1^). In addition, 100 mg L^−1^ of Fe^3+^ or Cr^6+^ resulted in an 87% or 99% FL quenching of the CQDs, respectively, indicating that the detection capacity of the CQDs to Fe^3+^ was also higher than Cr^6+^ (Appendix A). The coexisted HAs resulted in an overestimation of the concentration of Cr^6+^ in the solution (Figure 3B). As shown in Appendix A, 1 mg L^−1^ Cr^6+^ did not significantly reduce the FL intensity of the CQDs, while significant FL quenching of CQDs was observed in the solution with HAs coexisting. Correspondingly, with the HAs coexisting, the LOD of the CQDs to Cr^6+^ decreased to 0.4692 mg L^−1^, and the R^2^ value between the *F*_0_/*F* and Cr^6+^ reduced from 0.981 to 0.962, indicating that HAs interfered with the stable detection of Cr^6+^ by the cyanobacteria CQDs and a lower concentration of Cr^6+^ could be detected by the CQDs with HAs coexisting.

### 3.4. The Mechanisms That HAs Interfered with the Detection of Fe^3+^ and Cr^6+^ Ions Using CQDs

The coexisting HAs interacted with CQDs and changed the properties of CQDs. Specifically, with the coexistence of HAs, there is a new absorption peak at 320 nm (Figure 4A), indicating an n→π* electronic transition of the C=O groups on the CQDs [22]. This is because there are many functional groups, including –OH, –C=O, and –C–O, on the HAs [28], facilitating their interaction with CQDs. Results from the FTIR spectrum also proved this interaction. As shown in Figure 4B, the HAs coexisting resulted in an obvious red shift of –C–H from 3056.9 to 3066.8 cm^−1^ and –C=C from 1653.2 to 1659.7 cm^−1^, indicating a chemical reaction between CQDs and HAs. Moreover, the intensity of the –C–H and –C=C peaks was reduced while that of –C=O peaks was enhanced (Figure 4B), indicating potential oxidation of CQDs by HAs. The abundant N-doped functional groups led to strong reducibility of the cyanobacteria CQDs, and charge transfer from the surface of CQDs to HAs resulted in FL quenching of the CQDs [23,48]. Results from the XPS spectrum further proved the oxidation of CQDs by HAs (Figure 4C–F). Specifically, the HAs coexisting resulted in two new peaks corresponding to O=C–O at 287.9 eV and O=C–O at 530.9 eV, and the proportion of C–C/C=C structures reduced from 28.86 to 14.69%, while that of C=O structure increased from 9.47 to 27.31% in the complexation of CQDs and HAs (Figure 4D), both of which were indicative of the enhanced oxidation of the CQDs. The results from the FTIR and XPS spectrum suggested that HAs interacted with the CQDs, resulting in changes in CQDs structure and FL properties, which would also further affect the detection of metal ions by the cyanobacteria CQDs.

The TEM results suggested that HAs promoted the reunification of CQDs, affecting the dispersion of CQDs (Appendix A). On the one hand, the reunited CQDs by HAs may form steric hindrance between CQDs and metal ions, decreasing their accessibility. On the other hand, HAs may act as electron carriers to promote the contact of CQDs and metal ions [48,49], which would result in the overestimation of CQDs to the concentration of metal ions. In this study, we found that the coexisted HAs affected the detection of Fe^3+^ and Cr^6+^ by the cyanobacteria CQDs differently (Figure 3). It was because (i) the mechanisms resulting in FL quenching of the CQDs by Fe^3+^ and Cr^6+^ were different, and (ii) interaction between the HAs and the two metal ions was different.

It is well known that interaction between CQDs and metal ions can be ascribed to the abundant functional groups, including –C=C, –C=O, –O–H, and –N–H, on the CQDs (Figure 2). The dominant IFE effect between CQDs and Fe^3+^ or Cr^6+^ ions suggested that the static FL quenching phenomenon of the CQDs by the two metal ions may be because they consumed energy released from the excited electrons and thus relaxed the electrons to a ground state [41]. The ground-state complexation of CQDs and metal ions was generally non-luminous [50]. This also generally results in non-radiative recombination of the energy on the CQD surface [50]. As for the detection of CQDs to Fe^3+^ ions, only IFE occurred. The HAs coexisting wrapped the CQDs to form a barrier between the CQDs and Fe^3+^ ions (Appendix A) and absorbed the dissociated Fe^3+^ ions, decreasing their quenching effects on the FL intensity of CQDs [51]. Moreover, the oxidation of CQDs by HAs consumed electrons on the surface of CQDs (Figure 1 and Figure 4); competition of HAs and Fe^3+^ on the surface electrons inhibited the FL quenching of the CQDs by Fe^3+^. All the above findings lead to the underestimation of Fe^3+^ ions during CQD detection (Figure 5). Regarding the detection of CQDs to Cr^6+^ ions, both IFE and FRET occurred. A close distance between the energy donor and acceptor is one of the prerequisites for the FL quenching of CQDs by Cr^6+^ based on a FRET mechanism [43,52]. The HAs coexisting could act as a carrier to enhance the electron transfer between CQDs and Cr^6+^ ions, enhancing FRET (Figure 2 and Figure 5). Moreover, the dissolved HAs can highly interact with Cr^6+^ ions [37], leading to the transfer of the fluorophore and the resultant FL quenching. Finally, the coexisting HAs resulted in the overestimation of Cr^6+^ ions during CQD detection (Figure 3 and Figure 5).

## 4. Conclusions

Cyanobacteria CQDs could be used to detect Fe^3+^ and Cr^6+^ in aqueous environments. Compared to Cr^6+^, the CQDs showed higher sensitivity, detection capacity, and a wider linear detection for Fe^3+^. A redox reaction between the Fe^3+^ and functional groups on the CQDs resulted in FL quenching of the CQDs by Fe^3+^ based on the IFE mechanism. Apart from the IFE mechanism, there were FRET effects between the CQDs and Cr^6+^. During the detection of the metal ions by CQDs, the coexisted HAs oxidized and wrapped the CQDs, resulting in an underestimation of Fe^3+^ but an overestimation of Cr^6+^ by the CQDs. The interference of the HAs on the detection of metal ions by the CQDs sensor could not be ignored during the implication of CQDs in a natural aqueous environment.

## Figures and Tables

**Figure 1 ijerph-19-10225-f001:**
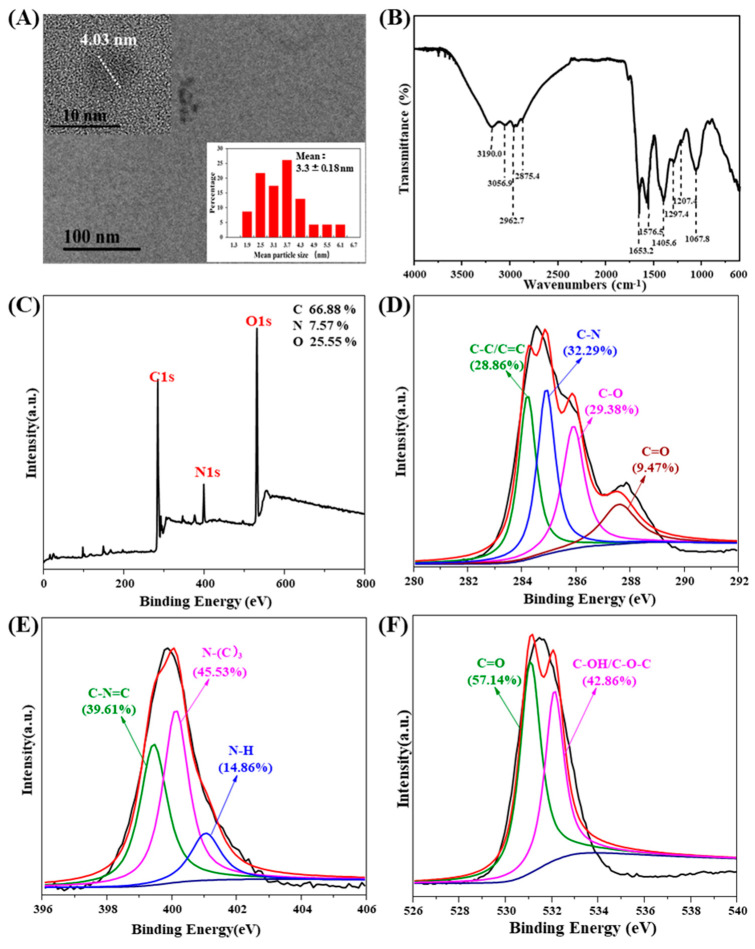
Morphological and spectroscopy analysis of the cyanobacteria carbon quantum dots (CQDs). (**A**) Transmission electron microscope image and the particle size distribution; (**B**) Fourier-transform infrared spectrum; (**C**) X-ray photoelectron spectroscopy (XPS) full-survey spectrum; (**D**) XPS high-resolution scan of C1s; (**E**) XPS high-resolution scan of N1s; (**F**) XPS high-resolution scan of O1s.

**Figure 2 ijerph-19-10225-f002:**
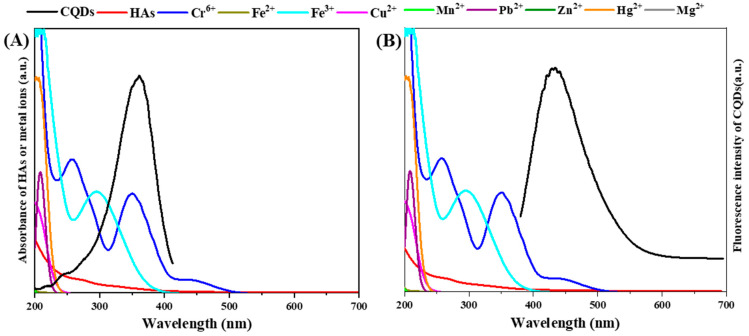
(**A**) The overlaps of the excitation spectrum of the cyanobacteria carbon quantum dots (CQDs) from 200 to 400 nm with emission wavelength at 430 nm with the absorption spectrum of humic acids (HAs) and metal ions. (**B**) The overlaps of the emission spectrum of CQDs from 380 to 700 nm with excitation wavelength at 360 nm with the absorption of HAs and metal ions.

**Figure 3 ijerph-19-10225-f003:**
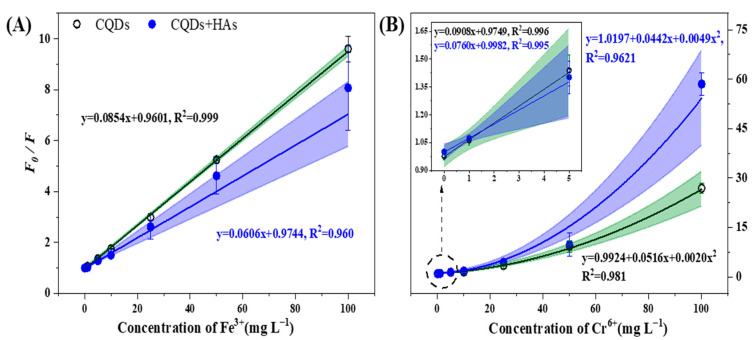
Fluorescence quenching of the cyanobacteria carbon quantum dots (CQDs) by (**A**) Fe^3+^ and (**B**) Cr^6+^ as affected by humic acids (HAs). *F*_0_/*F* indicates the initial fluorescence intensity of CQDs divided by the fluorescence intensity of CQDs after interaction with Fe^3+^ or Cr^6+^ ions. The shaded area represents the confidence bands of the data, and the error bars represent standard deviations (n = 3).

**Figure 4 ijerph-19-10225-f004:**
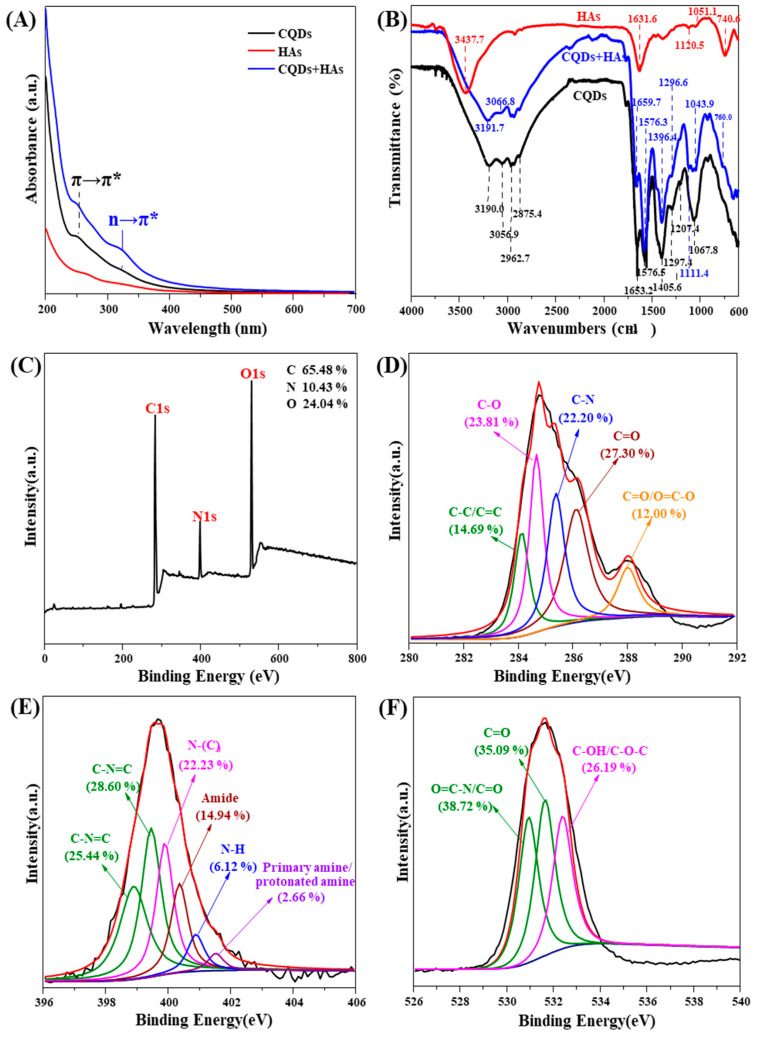
(**A**) Ultraviolet-visible (UV-vis) absorption spectra of the cyanobacteria carbon quantum dots (CQDs) as affected by humic acids (HAs); (**B**) Fourier-transform infrared spectrum of the CQDs as affected by HAs; (**C**) X-ray photoelectron spectroscopy (XPS) full-survey spectrum of the CQDs with the interaction of HAs; (**D**) XPS high-resolution scan of the C1s on CQDs with the interaction of HAs; (**E**) XPS high-resolution scan of the N1s on CQDs with the interaction of HAs; (**F**) XPS high-resolution scan of the O1s on CQDs with the interaction of HAs.

**Figure 5 ijerph-19-10225-f005:**
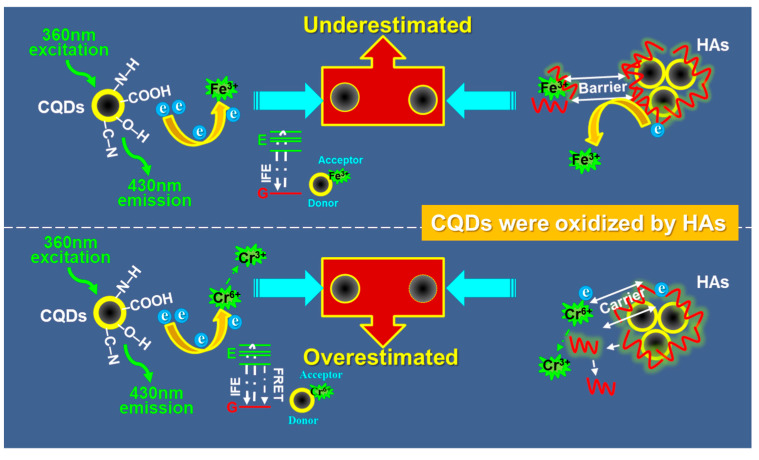
Schematic diagram of the mechanisms that humic acids (HAs) interfered with the detection of Fe^3+^ and Cr^6+^ ions by the cyanobacteria carbon quantum dots (CQDs). E—Excited-state, G—Ground-state, IFE—internal filtering effects, FRET—fluorescence resonance energy transfer.

## Data Availability

All sources were provided in the paper.

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
