# Peer review of "Humic Acids Affect the Detection of Metal Ions by Cyanobacteria Carbon Quantum Dots Differently"

_ijerph, 2022, doi:10.3390/ijerph191610225_

Round 1
Reviewer 1 Report
Simin Liu et al, reported a research manuscript titled "Humic acids affect the detection of metal ions by cyanobacteria carbon quantum dots differently". Authors synthesized carbon quantum dots from cyanobacteria, characterized and evaluated for their metal ion detection ability in presence of humic acids. The synthesized cyanobacteria carbon quantum dots are capable of detecting Fe3+ and Cr6+ metal ions in aqueous environments. This research article fit in the aim and scope of IJERPH. However, this article has the scope for further improvement upon revisions.
Comments:
1. Figure 1A: The particle shape, size and surface morphology was not clear from this TEM image. Since the size of CQDs were very small, reviewer suggest authors to include a high resolution TEM image to support the surface morphology of the CQDs.
2. Section 3.2: from line 260-267. This is a very general paragraph talking about metal ion detection importance. Reviewer suggest authors on include this paragraph in the introduction rather than in results & discussion. It should be more appropriate to discuss the corresponding results in this section.
3. Line 62: is it absorption or adsorption spectrum?
Author Response
REVIEWER #1:
Simin Liu et al, reported a research manuscript titled "Humic acids affect the detection of metal ions by cyanobacteria carbon quantum dots differently". Authors synthesized carbon quantum dots from cyanobacteria, characterized and evaluated for their metal ion detection ability in presence of humic acids. The synthesized cyanobacteria carbon quantum dots are capable of detecting Fe3+ and Cr6+ metal ions in aqueous environments. This research article fit in the aim and scope of IJERPH. However, this article has the scope for further improvement upon revisions.
Response: Thanks. We have addressed all your comments and responded to them point by point as described below.
Comment 1: Figure 1A: The particle shape, size and surface morphology was not clear from this TEM image. Since the size of CQDs were very small, reviewer suggest authors to include a high resolution TEM image to support the surface morphology of the CQDs.
Response: Thanks for your suggestion. A high-resolution TEM images has been supplied to support the surface morphology of the CQDs. (See Figure 1A in the revised manuscript)
Comment 2: Section 3.2: from line 260-267. This is a very general paragraph talking about metal ion detection importance. Reviewer suggest authors on include this paragraph in the introduction rather than in results & discussion. It should be more appropriate to discuss the corresponding results in this section.
Response: Thanks for this constructive comment. The introduction section and section 3.2 has been revised accordingly:
“Mental ions such as Cr6+ and Fe3+ are industrial products to be introduced into water environments via rain off or runoff (Liu et al. 2018; Zuo et al. 2018). Exposure to metal ions poses heavy threats to environmental health (Ananthanarayanan et al. 2014; Sanders et al. 1998; Shanker and Venkateswarlu 2011). It was reported that Cr6+ exposure led to significantly shorten the lifespan of fish and increased risks of respiratory diseases and lung cancer for human beings (Prasad et al. 2021).” (See Lines 57~63 in the revised manuscript)
Comment 3: Line 62: is it absorption or adsorption spectrum?
Response: Thanks. It should be absorption spectrum. (See Line 72 in the revised manuscript).

Reviewer 2 Report
The authors synthesized carbon quantum dots (CQD) using cyanobacteria for the humic acid affects on detection of three metal ions. The manuscript contains some novel information and the results are helpful for the scientific community. However I have some minor concerns before further process of Manuscript. The language of manuscript should be checked carefully, there found some grammatical mistakes. The quality of Fig.3 a&b should be improved. Authors mentioned various ions but mostly in figures they consider Cr and Fe, other ions should also compared at least in Fig 2.
Author Response
REVIEWER #2:
The authors synthesized carbon quantum dots (CQD) using cyanobacteria for the humic acid affects on detection of three metal ions. The manuscript contains some novel information and the results are helpful for the scientific community. However, I have some minor concerns before further process of Manuscript.
Response: We thank the reviewer for his/her constructive comments. We have addressed all of the reviewer’s comments and responded to them point by point as described below.
Comment 1: The language of manuscript should be checked carefully, there found some grammatical mistakes.
Response: Thanks. The spelling and grammar have been checked and improved throughout the revised manuscript. For example:
“Cyanobacteria are one of the main culprits of eutrophication, a serious environmental issue in aqueous environments.” (See Lines 42~43)
“Therefore, the coexisted HAs are likely to be a bottleneck for accurate FL detection of metal ions by CQDs technology in natural water.” (See Lines 92~93)
“We hypothesize that abundant N-doped surface functional groups could be formed on the surface of CQDs, contributing to their efficient detection of metal ions” (See Lines 98~100)
“To determine effects of HAs on FL intensity of the CQDs with the excitation at 360 nm….” (See Line 160)
“The C=C infers a graphene structure of the CQDs (Zhang et al., 2018).” (See Line 228)
“The FL intensity of CQDs against wavelength in the solution contaminated with Fe3+ or Cr6+ suggested that FL quenching degree of the CQDs increased with the increases in concentration of Fe3+ (Fig. S3) and Cr6+ (Fig. S4).” (See Lines 334~336)
“The coexisting HAs interacted with CQDs and changed the properties of CQDs. Specifically, with the coexistence of HAs, there is a new absorption peak at 320 nm (Fig. 4A), indicating an n→π* electronic transition of the C=O groups on the CQDs (Mondal et al., 2019).” (See Lines 385~388)
“Results from FTIR spectrum also proved this interaction.” (See Lines 390~391)
“Moreover, oxidation of CQDs by HAs would consume electrons on the surface of CQDs (Figs. 1 and 4); competition of HAs and Fe3+ on the surface electrons inhibited the FL quenching of the CQDs by Fe3+.” (See Lines 429~431)
“Compared to Cr6+, the CQDs showed higher sensitivity, detection capacity and a wider linear detection for Fe3+. Redox reaction between Fe3+ and functional groups on the CQDs resulted in FL quenching of the CQDs by Fe3+ based on IFE mechanism.” (See Lines 454~457)
Comment 2: The quality of Fig.3 a&b should be improved.
Response: Thanks for your comments. We’ve improved the quality of Fig. 3 and rearranged this figure in the revised manuscript.
Comment 3: Authors mentioned various ions but mostly in figures they consider Cr and Fe, other ions should also compare at least in Fig 2.
Response: Thanks for this constructive comment. The associated discussion has been added in the revised manuscript as follows:
“As shown in Fig. 2A, the excitation spectra of CQDs specifically overlap with the absorption spectra of Cr6+ and Fe3+ in a large area, while only a small overlap with the absorption spectra of Hg2+, Cu2+ and Pb2+, and none overlap with the absorption spectra of Fe2+, Mn2+, Zn2+ and Mg2+. This suggests that the cyanobacteria CQDs are sensitive to the detection of Cr6+ and Fe3+ ions, showing significant FL quenching of the CQDs by the two ions with an IEF mechanism. The Cr6+ showed broad absorption peaks at 260 and 350 nm, and Fe3+ showed an absorption peak at 300 nm. Except for Cr6+, the absorption spectra of the other eight ions do not overlap with the emission spectra of CQDs (Fig. 2B). This suggests that the cyanobacteria CQDs could also interact with Cr6+ and result in FL quenching of CQDs based on a FRET mechanism. The metal ions with broad absorption spectra and high extinction coefficients are generally acted as FL quenchers for FRET acceptors (Zhang et al., 2019).” (See Lines 287~298)

Reviewer 3 Report
Finding novel and green fluorescent materials has been a popular research area for a while. The authors investigated a new way to sensitively detect metal ions using carbon quantum dots (CQDs) fabricated from cyanobacteria. The surface characterization of the synthesized CQDs demonstrated the new materials were N rich. In addition, the authors applied the synthesized CQDs to the detection of metal ions under different conditions, such as pH and with/without the presence of humic acids. Overall, the conclusions were supported by the experimental results. There are a few areas that the authors should pay attention.
1. Moderate English editing. There are many grammar mistakes throughout the manuscript. Just to name a few, line 40, 44, 62 etc. It is strongly recommended the authors ask a professional or native speaker to review their manuscript.
2. Line 226. It would be nice to have a discussion why pH matters in FL, particularly for this manuscript.
3. Line 288. The authors claimed the detection of Fe3+ by CQDS is precise, accurate, and stable. None of the provided data supported this statement. Precision is about the reproducibility of the data; accuracy is about how close the measurement is to the true value, and stability data were not provided. The sole comparison between two LODs did not support the statement.
4. Line 344-5. The authors proposed the difference in FL quenching mechanism and interactions with humic acids between Fe3+ and Cr3+, and emphasized the enhanced FRET by Cr3+. Is there any theory to predict this FRET effect? How about metal ion structure or the complex structure? If a new metal ion is used, how could you know which quenching route it would take?
Author Response
REVIEWER #3:
Finding novel and green fluorescent materials has been a popular research area for a while. The authors investigated a new way to sensitively detect metal ions using carbon quantum dots (CQDs) fabricated from cyanobacteria. The surface characterization of the synthesized CQDs demonstrated the new materials were N rich. In addition, the authors applied the synthesized CQDs to the detection of metal ions under different conditions, such as pH and with/without the presence of humic acids. Overall, the conclusions were supported by the experimental results. There are a few areas that the authors should pay attention.
Response: Thanks. We have addressed all your comments and responded to them point by point as described below.
Comment 1: Moderate English editing. There are many grammar mistakes throughout the manuscript. Just to name a few, line 40, 44, 62 etc. It is strongly recommended the authors ask a professional or native speaker to review their manuscript.
Response: Thanks for your comments. The spelling and grammar have been checked and improved throughout the revised manuscript by a professional English speaker. Some language revisions in the revised manuscript are shown as follows:
“A “top-down” method for hydrothermal synthesis of CQDs is both cost-effective and environmentally friendly.” (See Lines 40~41)
“Cyanobacteria are one of the main culprits of eutrophication, a serious environmental issue in aqueous environments. Development of feasible methods for recycle and reuse widespread cyanobacteria biomass is critical to addressing this issue” (See Lines 42~44)
“Fe2+ interacted with the –OH, –NH2, –COOH groups on the surface of CQDs derived from mango leaves, resulting in a decreased FL intensity of the CQDs and the decreased FL intensity is in line with the concentration of Fe2+ (Singh et al., 2020).” (See Lines 68~71)
“Therefore, the coexisted HAs are likely to be a bottleneck for accurate FL detection of metal ions by CQDs technology in natural water.” (See Lines 92~93)
“We hypothesize that abundant N-doped surface functional groups could be formed on the surface of CQDs, contributing to their efficient detection of metal ions” (See Lines 98~100)
“To determine effects of HAs on FL intensity of the CQDs with the excitation at 360 nm….” (See Line 160)
“The C=C infers a graphene structure of the CQDs (Zhang et al., 2018).” (See Line 228)
“The FL intensity of CQDs against wavelength in the solution contaminated with Fe3+ or Cr6+ suggested that FL quenching degree of the CQDs increased with the increases in concentration of Fe3+ (Fig. S3) and Cr6+ (Fig. S4).” (See Lines 334~336)
“The coexisting HAs interacted with CQDs and changed the properties of CQDs. Specifically, with the coexistence of HAs, there is a new absorption peak at 320 nm (Fig. 4A), indicating an n→π* electronic transition of the C=O groups on the CQDs (Mondal et al., 2019).” (See Lines 385~388)
“Results from FTIR spectrum also proved this interaction.” (See Lines 390~391)
“Moreover, oxidation of CQDs by HAs would consume electrons on the surface of CQDs (Figs. 1 and 4); competition of HAs and Fe3+ on the surface electrons inhibited the FL quenching of the CQDs by Fe3+.” (See Lines 429~431)
“Compared to Cr6+, the CQDs showed higher sensitivity, detection capacity and a wider linear detection for Fe3+. Redox reaction between Fe3+ and functional groups on the CQDs resulted in FL quenching of the CQDs by Fe3+ based on IFE mechanism.” (See Lines 454~457)
Comment 2: Line 226. It would be nice to have a discussion why pH matters in FL, particularly for this manuscript.
Response: Thanks for this constructive comment. How and why pH affect fluorescence intensity of CQDs has been discussed in the revised manuscript as follows:
“Effects of pH on FL intensity of CQDs can be ascribed to protonation and deprotonation of functional groups such as C=O, O–H, and various N-doped functional groups on the surface of CQDs (Huang et al., 2022; Yang et al., 2020; Li et al., 2018). In alkaline conditions, the fluorescence intensity of CQDs can be ascribed to oxidation of C-O functional groups (Chai et al., 2015).” (See Lines 262~269)
References:
Huang et al. (2022) Carbon dots derived from Poria cocos polysaccharide as an effective “on-off” fluorescence sensor for chromium (VI) detection. Journal of Pharmaceutical Analysis. 12:104-112
Yang et al. (2020) A novel and sensitive ratio metric fluorescence assay for carbendazim based on N-doped carbon quantum dots and gold nanocluster nanohybrid. Journal of Hazardous Materials. 386:121958
Li et al. (2018) High quantum yield nitrogen-doped carbon dots: green synthesis and application as “off-on” fluorescent sensors for the determination of Fe3+ and adenosine triphosphate in biological samples. Sensor Actuator B Chem. 276:82-88
Chai et al. (2015) Functionalized carbon quantum dots with dopamine for tyrosinase activity monitoring and inhibitor screening: in vitro and intracellular investigation. ACS applied materials & interfaces. 7:23564-23574
Comment 3: Line 288. The authors claimed the detection of Fe3+ by CQDS is precise, accurate, and stable. None of the provided data supported this statement. Precision is about the reproducibility of the data; accuracy is about how close the measurement is to the true value, and stability data were not provided. The sole comparison between two LODs did not support the statement.
Response: Thanks, this sentence has been deleted in the revised manuscript.
Comment 4: Line 344-5. The authors proposed the difference in FL quenching mechanism and interactions with humic acids between Fe3+ and Cr6+, and emphasized the enhanced FRET by Cr6+. Is there any theory to predict this FRET effect? How about metal ion structure or the complex structure? If a new metal ion is used, how could you know which quenching route it would take?
Response: Thanks for the reviewer’s questions.
- Yes, the theory of fluorescence resonance energy transfer (FRET) effect is a non-radiative energy transfer process (Neema et al. 2020). Two necessary conditions for the FRET process to occur include on one hand, distance between energy donor and acceptor should less than 10 nm, and on the other hand, there is a large area of overlap between the emission spectrum of the energy donors and the absorption spectrum of the energy acceptor (Neema et al. 2020; Sapsford et al. 2006).
- It was reported that CQD can interact with metal ions and pesticides via a FRET mechanism (Gholami et al. 2020). Generally, a substance with broad absorption spectra and high extinction coefficients can act as FL quenchers and result in FL quenching of CQDs via FRET effects (Zhang et al. 2019). In this study, FRET effects occurred between CQDs and Cr6+, which may because Cr6+ have a broad absorption spectrum, which overlapped with the emission spectrum of CQDs in a large area. Interaction between CQDs and specific metal ions depends on the raw materials of CQDs and the metal ion species, which is because the structural properties of CQDs derived from different raw materials are highly diverse and the absorption spectrum of different metal ions are highly different (Molaei 2020; Yang et al. 2018). We also added some discussion to address this comment in the revised manuscript as follow: “The Cr6+ showed broad absorption peaks at 260 and 350 nm, and Fe3+ showed an absorption peak at 300 nm. Except for Cr6+, the absorption spectra of the other eight ions do not overlap with the emission spectra of CQDs (Fig. 2B). This suggests that the cyanobacteria CQDs could also interact with Cr6+ and result in FL quenching of CQDs based on a FRET mechanism. The metal ions with broad absorption spectra and high extinction coefficients are generally acted as FL quenchers for FRET acceptors (Zhang et al., 2019).” (See Lines 292~298)
- To determine whether a new metal ion can result in FL quenching of the CQDs with FRET mechanisms, we need to first detect whether the absorption spectrum of the new ions can overlap with the emission spectrum of CQDs with a large area. In addition, if the specific functional groups on the surface of CQDs and the structure characters of metal ions have been determined, theoretical calculation methods such as density functional theory method might be used to explore the potential of molecular interactions between metal ions and CQDs as well as the resultant FL quenching of CQDs (Tang et al. 2020).
- In this study, the abundant functional groups on surface of the CQDs have been determined using FTIR analysis. Therefore, in our future works, we’ll predict the potential interaction routes between different metal ions and the cyanobacteria CQDs by combining with theoretical calculation methods. Thanks for your constructive suggestion.
References:
Neema et al. (2020) Chemical sensor platforms based on fluorescence resonance energy transfer (FRET) and 2D materials. Trends Analyt Chem 124:115797
Sapsford et al. (2006) Materials for fluorescence resonance energy transfer analysis: beyond traditional donor–acceptor combinations. Angewandte Chemie International Edition 45:4562-4589
Gholami et al. (2020) A new nano biosensor for maitotoxin with high sensitivity and selectivity based fluorescence resonance energy transfer between carbon quantum dots and gold nanoparticles. Journal of Photochemistry and Photobiology A: Chemistry 398:112523
Molaei (2020) Principles, mechanisms, and application of carbon quantum dots in sensors: a review. Analytical Methods 12:1266-1287
Tang et al. (2020) Monitoring graphene oxide’s efficiency for removing Re (VII) and Cr (VI) with fluorescent silica hydrogels. Environmental Pollution 262:114246
Yang et al. (2018) Controllable and eco-friendly synthesis of P-riched carbon quantum dots and its application for copper (II) ion sensing. Applied Surface Science 448:589-598
